# Daily Living Subjective Cognitive Decline Indicators in Older Adults with Depressive Symptoms: A Scoping Review and Categorization Using Classification of Functioning, Disability, and Health (ICF)

**DOI:** 10.3390/healthcare10081508

**Published:** 2022-08-10

**Authors:** Renata Komalasari, Elias Mpofu, Gayle Prybutok, Stanley Ingman

**Affiliations:** 1Department of Rehabilitation and Health Services, University of North Texas, Chilton Hall, 410 Avenue C, Suite 289, Denton, TX 76201, USA; 2School of Health Sciences, University of Sydney, Camperdown, NSW 2050, Australia; 3School of Human and Community Development, University of Witwatersrand, Johannesburg 2000, South Africa

**Keywords:** subjective cognitive decline indicators, activities of daily living, instrumental activities of daily living, depressive symptoms, International Classification of Functioning, Disability, and Health, Bayer activities of daily living scale, Lawton-Brody IADL scale, Katz index of independence

## Abstract

(1) Background: This scoping review identifies subjective cognitive decline (SCD) indicators in ADLs and instrumental activities of daily living (IADLs) in older adults with depressive symptoms using the WHO International Classification of Functioning, Disability, and Health (ICF). (2) Methods: We searched Medline via Ebscohost, Pubmed, and PsycINFO for articles published on activities of daily living (ADL) indicators of SCD in older adults with depressive symptoms, published in English language journals from January 2011 to November 2021. Following the flow diagram, 2032 titles and abstracts were screened for relevance based on the Population, Concept, and Context inclusion and exclusion criteria. (3) Results: Eight articles provided evidence about the ADL indicators of SCD in older adults with depressive symptoms. The analysis yielded indicators based on low and high cognitively demanding tasks assessed on five different scales. Framed on the ICF categorization and coding system, the SCD-ADL indicators are personal care, mobility, and general tasks and demands; SCD-IADL indicators are mobility, general tasks and demands, learning and applying knowledge, domestic life, communication, major life areas, and community, social, and civic life. (4) Conclusion: Highly cognitively demanding activities present more difficulties for individuals with SCD, making IADLs a stronger predictor of SCD than ADLs.

## 1. Introduction

Subjective cognitive decline (SCD) is an individual’s perceived concerns about a memory decline compared to previous cognitive function in the absence of objectively measured cognitive deficits [1]. If these worries are chronic, they might be associated with depressive symptoms from a loss of a sense of mental presence impacting activities of daily living (ADLs). Older adults with SCD may notice the incipient presence of depression in how they perform activities of daily living they used to do but now do less well than before [1], reducing their quality of life. The underlying reasons are that the experience of SCD is associated with prevalent mood disorders that may be part of the mental function loss experience [2] or a reaction to the frustration of losing functional abilities [3]. Individuals with greater SCD may be at an increased risk for further objective declines in cognitive function like mild cognitive impairment (MCI) and Alzheimer’s disease (AD) [4]. Therefore, it is essential to identify those at risk for MCI or AD and intervene as early as possible [3,4]. A higher degree of depressive symptoms is strongly associated with reduced cognitive function and functional decline in people with MCI or AD [5]. SCD may potentially indicate a preclinical MCI/AD stage. Therefore, the factors contributing to the risk of progression from MCI to dementia should be further investigated, taking into account other environmental factors, such as social support and nutritional status, that may magnify those risks [6].

Behavioral and cognitive study findings showed that deficits in everyday activities noted earlier in the progression of dementia provide a better representation of cognitive impairment [7,8,9]. While subtle decrements associated with age are expected, individuals with SCD have more severe difficulties in everyday functioning [7,10], which may deteriorate into Alzheimer’s disease (AD) over time [5]. The disease process related to progression to MCI or AD impacts basic and instrumental ADLs, which are tasks required to function daily [11].

### 1.1. ADLs and IADLs 

Activities of daily living (ADL). ADLs include core tasks of everyday life mainly to maintain personal care, such as bathing, toileting, eating, and grooming [11]. More recent literature associates basic ADLs with low cognitively demanding tasks, including daily routine, transferring, and preparing medications, distinct from high cognitively demanding ones [5,8]. Older study findings showed no significant association between cognitive function and basic self-care skills amongst individuals with moderate AD [12]. However, more recent evidence indicates impairments in ADLs, in general, strongly predict progression to dementia in patients in prodromal stages of AD [5,8].

### 1.2. Instrumental Activities of Daily Living (IADLs)

IADLs are more complicated than the basic activities of daily living to achieve independent living (Wallace, 2007). These activities require more cognitive resources or higher neuropsychological functioning than simple or basic ADLs [11,13]. More cognitively demanding activities, including making a phone call or handling finances, are more vulnerable to early cognitive changes than activities with low cognitive demands such as personal hygiene or preparing food [8,14]. Hence, highly cognitively demanding IADLs are an early marker of MCI [8]. An increasing volume of research using everyday functioning as a cognitive decline indicator in people with MCI tend to focus on deficits in IADLs rather than in ADLs [5,13]. Subtle IADL deficits in MCI may be well detected if a reliable identification of performance-based assessment instruments is available to facilitate a valid and reliable assessment [13].

The use of assessment tools specifically designed and validated for patients with SCD is strongly recommended. An example of an everyday functioning assessment scale that includes ADLs and IADLs is the Bayer-ADL (B-ADL) scale [15]. Based on a factor analysis, the B-ADL subdivided everyday tasks into items with lower cognitive demand (ADLs) and higher cognitive demand (IADLs) [14,15]. The International Classification of Functioning, Disability, and Health (ICF) frames activity and participation with a coding system to aid the organization and interpretation of study [16]. Surprisingly, the ICF framework does not provide a clear line between ADLs and IADLs, challenging differentiation between the two categories. Some phenotypes listed in the ICF distinctively belong to the IADL category due to their complex nature requiring higher cognitive processes. These tasks were learning and applying knowledge, communication, major life areas, community, social and civic life, and problems encountered in social and community life involvement. However, phenotypes, such as mobility and general tasks and demands, may well apply to either ADLs or IADLs, depending on the complexity level of the task, again posing a challenge for categorization. The evidence for the ADL and IADL indicators with SCD is yet to be aggregated to inform prospective studies and to develop screening measures for SCD in older adults, particularly those experiencing depressive symptoms.

### 1.3. Cognition in ADLs and IADLs 

Limitations in ADLs/IADLs are a strong predictor of cognitive decline [5,8] both in an individual with or without symptoms of cognitive impairment [17]. Evidence showed that IADLs were the first to decline, given the multiple cognitive resources and neuropsychiatric factors used in more advanced ADLs than in basic ADLs [18]. However, a more recent study suggested the importance of including both categories of activity performance in detecting a cognitive decline. The authors developed a new assessment tool to measure older adults’ everyday functioning in a geriatric day hospital in Brussels, Belgium, to distinguish cognitively healthy aging from pathologically cognitive aging among older adults with healthy cognition, MCI, and AD [19]. They adapted the Katz index for the basic ADL items and the Lawton Scale Instrumental IADLs for the IADL items [19]. They then compared the items of both ADL categories to the ICF-based terminology and scoring system. To determine the extent to which cognitive decline causes limitations in daily functioning, they calculated a disability index for global, cognitive, and physical functioning [19]. Based on the validity and reliability of the newly adapted assessment tool, ADLs and IADLs both have high accuracy to discriminate those with and without cognitive decline, validating the importance of assessing SCD-ADLs/IADLs indicators across the cognitive function continuum [19].

Progression to disability due to declined daily functioning has been used as a core criterion for dementia diagnosis [5,14,20]. Likewise, improved activity performance has indicated successful cognitive rehabilitation programs amongst older people with cognitive concerns [21]. However, there are challenges in using ADLs as indicators of cognitive decline. While relatively easy to observe, changes in everyday activities are often assumed as physiologic, age-related phenomena [14]. Moreover, a decline in daily performance occurs not only in the initial stage of dementia but rather across the cognitive aging continuum (normal cognition—SCD—MCI—AD) [5], challenging ADLs/IADLs determining SCD. In addition, the interplay of individual resources (i.e., access to healthcare) and sociodemographic factors (i.e., education or income levels) contributing to SCD may obscure the identification of restrictions in ADLs/IADLs [5]. In depressive comorbidities, older people with SCD may experience more difficulties maintaining ADLs [5], adding to the challenges in determining which ADL/IADL indicators predict SCD.

### 1.4. Depression and SCD

The experience of SCD is often comorbid with depression [9,22], with later-life depression seen as a prodrome of dementia [23]. However, study findings in this area have been inconclusive. A few large-scale longitudinal studies looking at the experience of early-stage cognitive decline in the presence of depressive symptoms showed that depression was not a predictor of MCI [6,8,24]. However, other studies showed the opposite [8,9]. Overlapping symptoms were reported in both MCI and depressive symptoms, including apathy, loss of interests and hobbies, trouble concentrating, impaired thinking, and social withdrawal [25]. Similar neurobiological changes in white matter may explain why individuals with either condition share a pattern of neuronal damage [23]. Given the confounding effect of depression, dementia diagnosis in a person with the early stage of dementia becomes challenging [5].

An increasing number of dementia studies have used a person’s ability to perform everyday activities to predict cognitive decline [5,8,9,26]. The neurological process of cognitive impairment occurs gradually [5], and it may take two to four years before a self-report restriction in activities of daily living [5,27]. The mechanism by which cognitive decline in the presence of depressive symptoms manifests in declined daily functioning remains unclear [18]. A cross-sectional study amongst 274 older adults in Brazil across the cognitive function levels revealed that participants with depression have lower performance than those who did not have depression in the cognitively healthy group but not in older adults with MCI, dementia, or AD [26]. This study suggests a progressive loss of association between depression and poorer cognition along the cognitive functioning continuum (cognitively healthy—MCI—dementia/AD), adding to the challenges of dementia diagnosis based on everyday functioning. This scoping review seeks to identify SCD indicators in ADLs and IADLs in older adults with depressive symptoms. Our specific research question was, “What is known from the existing literature about the indicators of SCD in ADLs and IADLs amongst older adults with depressive symptoms?” Better characterization of the ADL and IADL phenotypes is essential in recognizing early cognitive symptoms in people with SCD.

## 2. Materials and Methods

### 2.1. Research Design

A scoping review summarizes the emerging findings in a particular study area, identifying gaps in the existing knowledge for a more focused study [28]. A scoping review was appropriate for this study, given the absence of evidential synthesis on indicators of SCD in ADLs and IADLs important to successful aging. Successful aging is a dynamic process where one continues to learn and thrive despite old-age-related challenges [29].

#### 2.1.1. Search Strategy

We searched Medline via Ebscohost, Pubmed, and PsycINFO using the terms ‘activities of daily living,’ ‘older adults’, and ‘cognitive decline.’ After, we expanded our search terms to include those in Table 1. We collaborated with our research librarian, who recommended that we use Medline and Pubmed to retrieve peer-reviewed resources in the field of medicine and health, focused on subjective cognitive decline and activities of daily living. We also included PsycINFO, from which we retrieved articles on older adults with depressive symptoms.

#### 2.1.2. Inclusion/Exclusion Criteria

This stage involved selecting the articles in three steps: (1) title screening, (2) abstract screening, and (3) full article screening. As per inclusion and exclusion criteria, studies were selected based on the Population, Concept, and Context (PCC) framework [30]. Studies were selected if they: (1) included older adults with SCD (population), (2) described their activity of daily living (concept), and (3) included older adults experiencing depressive symptoms (context) (see Table 2). The search terms depression, depressed, and depressive symptoms were used, and articles that discussed depressive symptoms were part of the inclusion criteria for our search. Included studies were published from January 2011 to November 2021. We started the study in November 2021. We chose this date range to obtain an overview of the extent of research on older adults with subjective cognitive decline and activities of daily living. Studies were excluded if they were not on SCD in the activity of daily living amongst older adults with depressive symptoms or not in English. Studies were excluded if they were not on SCD in the activity of daily living amongst older adults with depressive symptoms or not in English.

#### 2.1.3. Summary of the Literature Search 

Figure 1 presents the summary of the search process and outcome. Beginning with 2032 records, we removed 498 duplicates, resulting in 1534 for eligibility screening. Further screening, using inclusion and exclusion criteria and records of abstracts, resulted in 28 papers for full-text assessment. Of these, only eight studies were included in the final analysis. 

#### 2.1.4. Characteristics of Studies

Eight studies were included in the final analysis, all of which were published articles [5,6,8,9,18,22,24,26]. Of these, four explored participants’ activities of daily living in people with early-stage cognitive decline or at high risk for dementia [6,8,9,22], while the remaining articles focused on depressive symptoms affecting the activities of daily living in older adults with cognitive decline or Alzheimer’s disease [5,18,24,26]. The total number of participants in the published articles was 4608, ranging from 36 [24] to 1386 [22], and distributed across two prospective studies [5,8] (*n* = 1697) and six quantitative cross-sectional or survey studies (*n* = 2911) [8,9,16,22,24,26] (See Table 3).

### 2.2. Data Synthesis

We utilized the WHO-ICF to guide our data synthesis. Following previous recommendations, the ICF provides a comprehensive analysis of experiences and needs from the person’s perspective [30]. This approach was that the ICF has codes for ADLs, IADLs, and SCD to aid the organization and interpretation of findings. After tracking all ADL/IADL assessment tools used in the published articles, we coded all the ADLs/IADLs assessed and then mapped them onto the ICF framework.

## 3. Results

### 3.1. Participants’ Characteristics

The mean age of participants ranged from 60 to 82.8 [5,8]. The percentage of female participants ranged from 44.14% (Yakhia et al., 2014) to 80.1% [9]. The participants’ years of education ranged from none to 17 years or university level. All published articles focused on older adults with cognitive decline and depressive symptoms. MCI was the most frequently studied condition among all other cognitive decline levels [8,9,18,22,24,26]. Only one study assessed older adults with SCD [5]. Three published articles included older adults with depressive symptoms as a comparative group sample [8,24,26]. In the remaining published articles, older adults with cognitive decline were being assessed for depressive symptoms for univariate analysis [5,8,9,18,22] (See Table 1).

### 3.2. SCD-ADL Indicators

Of the eight published articles being reviewed, only two assessed SCD-ADL indicators, measured on the adapted Katz Index of Independence scale [26] and the Bayer-Activities of Daily Living (B-ADL) [5,8]. The B-ADL referred to basic ADLs as low cognitively demanding activities. The SCD-ADL indicators consist of bathing, dressing, toileting, transferring, continence, and feeding [26], phenotyped as personal care on the ICF framework [16]. ADL indicators like transportation, shopping, and going for a walk without getting lost were phenotyped as mobility. Taking care of self, managing everyday activities, preparing food, using domestic appliances, and participating in leisure activities were phenotyped as general tasks and demands [26] (see Table 4).

The adapted Katz Index of Independence assessed the personal care activities in ADLs, which had expanded characteristics compared to the original Katz index [31]. For example, the adapted version had a score range of 0 to 10 [26], while the original Katz index had a score range of 0–6, with lower scores indicating more significant impairment [32]. The Katz index is a yes/no scale ranking adequacy of basic personal care, i.e., bathing, dressing, toileting, transferring, continence, and feeding [32]. Since its initial development [32], this scale has had different approaches to scoring; however, it has consistently demonstrated its utility in evaluating functional status in the elderly population [33]. The Bayer-ADL (B-ADL) scale subdivided everyday functionality into high and low cognitive demand [15]. The latter was used to identify ADL indicators of SCD (see Table 4).

Self-care. Self-care describes tasks about caring for oneself, consisting of washing oneself [26], caring for body parts, personal hygiene [5,8], toileting, dressing, eating, maintaining continence [26].General tasks. General tasks and demands describe tasks performed regularly. SCD indicators with this phenotype include managing daily routine, considered low cognitively demanding activity measured in the B-ADL scale [5,8].Mobility. Mobility consists of tasks like walking without getting lost [5,8], changing body position or self-transferring [24] and going for a walk without being lost [5,8].

### 3.3. SCD-IADL Indicators

Subjective cognitive decline-IADL (SCD-IADL) indicators are functional inadequacy in independent living skills that may signify SCD. All of the published articles included an assessment of SCD-IADLs [5,6,8,9,18,22,24,26]. Framed on the ICF [16], the SCD-IADL indicators were represented by eight phenotypes: general tasks and demands, mobility, domestic life, learning and applying knowledge, communication, major life areas, community, and social and civic life. The first two phenotypes, i.e., general tasks and demands and mobility, also served as the SCD-ADL indicators measured on the B-ADL [5,8] and adapted Katz index [32]. Unlike the SCD-ADL phenotype, SCD-IADL indicators represent high cognitively demanding activities. (See Table 4).

General tasks and demands. Two published articles assessed this phenotype, which included undertaking multiple tasks: continuing tasks after an interruption, doing two things at once, doing things safely [5,8], and handling stress and other psychological demands (performing tasks under pressure) [8].Mobility. Six published articles assessed this phenotype, including moving around, e.g., finding their way in an unfamiliar place [5,8], using transportation [5,6,8,18,22,26], driving or traveling out of neighborhood, driving, and arranging to take buses [18].Learning and applying knowledge. Five published articles included an assessment of this phenotype. The following are activities requiring learning and applying knowledge that indicate SCD: focusing attention (paying attention to, understanding, discussing TV, books, and magazines, and concentrating on reading) [5,8,18], thinking, writing, or keeping track of current events [18], and other specified learning and applications of knowledge such as literacy [9].Domestic life. All of the published articles assessed domestic life activities in assessing SCD. This phenotype consists of the acquisition of goods and services, such as shopping [5,8,9,18,22,26], acquisition of necessities [18], performing household tasks, such as preparing meals [5,6,8,9,18,22,24,26], doing housework (housekeeping, laundry) [18,22,25], and using household appliances, e.g., heating water, turning off the stove after use [5,8,18].Communication. Difficulties maintaining communication may signify SCD. Four published articles assessed this phenotype, including communicating with receiving spoken messages (giving direction, taking a message for someone else) [5,8], speaking or describing what was heard, giving directions when asked, taking part in a conversation, and using communication technology and devices (telephoning) [24,26].Interpersonal interactions and relationships. Limited interpersonal interactions and relationships may exacerbate cognitive decline. Only one published article assessed this phenotype, using the term interpersonal interactions and relationships, such as relating to a stranger [8].Major life areas. Seven published articles investigated major life areas as indicators of SCD, including education [9], writing checks, paying bills, balancing a checkbook [18], counting money, personal finances [5], handling money [5,6,8,9,18,22,26], and personal economic resources (assembling tax records, business affairs, or papers) [18].Community, social, and civic life. This phenotype was assessed in three published articles, where participants reported difficulties with community life. This phenotype included remembering appointments, family occasions, holidays, medications [18], recreation and participating in leisure and hobby activities [5,8], and other specified community, social and civic life activities [5].

### 3.4. SCD-ADL/IADL Assessment Scales

Five assessment scales were used in the published articles that measure everyday functionality in older adults: Bayer-ADLs (B-ADLs) [15], Functional Activities Questionnaire (FAQ) [34], Alzheimer’s Disease Cooperative Study ADL-Prevention Instrument (ADL-PI) [35], Lawton-Brody IADL [36], and ecological assessment of IADLs [24]. The B-ADL scale listed high cognitive demand activities as follows: coping with unfamiliar situations, performing a task when under pressure, describing what one has just seen or heard, continuing with the same task after a brief interruption, taking a message for someone else, observing important dates or events, doing two things at the same time, finding one’s way in an unfamiliar place, giving directions if asked the way, taking part in a conversation, and concentrating on reading [5,8]. Other IADL indicators of SCD were mobility, verbal learning, subcomponents of executive function measured on the ADL-PI [9], the ability to use a phone, shopping, food preparation, laundry, mode of transportation, responsibility for own medication, and managing personal finances measured on the Lawton IADL scale [6,22,26] (see Table 4).

These internationally based functional deficit assessment scales have good to excellent psychometric properties. For example, the Bayer-ADL scale, a 25-item valid indicator of cognitive status, has remarkable internal consistency (Cronbach α = 0.94) [37]. It has a score range of 0–30 [15], where higher scores correspond to a more severe functional deficit or dependence [15] and can discriminate older adults with early stages of dementia or cognitive impairment [15]. Another example, the Lawton IADL scale, an eight-domain function scale with a gender-specific score range, has excellent reliability (inter-rater reliability at 0.85.) [38]. It helps assess functioning at present to deterioration over time with a score range of 0 (low function, dependent) to 8 (high function, independent) for women and 0 through 5 for men, given the exclusion of food preparation, housekeeping, and doing laundry for assessment in men [38]. The ADL-PI has adequate reliability [35] and a score ranging from 0 to 45, where higher scores indicate more functional independence [9]. The FAQ items were reported as having the most accurate properties, with good sensitivity (85%) and high reliability (exceeding 0.90) [33] in discriminating between and predicting progression from cognitively normal individuals to MCI [39].

These SCD-IADL indicator assessment scales were used in various care settings with different administration modes. The Bayer-ADL [5,8], the ADL-PI [9], the adapted Lawton-Brody IADL [26], and the Lawton-Brody IADL [6,22] were administered to either an individual or a caregiver (spouse or informant sufficiently familiar with the individual) or both. The Lawton-Brody IADL scale was used in a memory clinic by referral [18]. Its usability was targeted for community or hospital settings but not for institutionalized older adults [38]. The FAQ, however, was used as a self-rated measure of IADLs to discriminate between dementia and non-dementia and between MCI and AD subjects [18]. The ADL-PI was completed by participants recruited from both community and primary care clinics [9], while the ecological assessment of IADL was used by recording participants’ activity with two monocular video cameras and observing the activities on an actigraphy in an observation room [24].

## 4. Discussions

This review has investigated the activities of daily living indicators of SCD among older adults with depressive symptoms, analyzing data from eight published articles. Our findings identified three basic ADL phenotypes indicating SCD: personal care, general demand and tasks, and mobility. Poorer self-report of everyday functioning has been associated with poorer executive functioning and temporal order memory [7]. The last two phenotypes (e.g., general demand and tasks and mobility) also function as SCD-IADL indicators. Similar to previous findings [14,37], we also identified more IADL than ADL phenotypes indicating SCD in older adults with depressive symptoms. Compared to basic ADLs, SCD was more well represented by deficits in IADLs.

Our study confirms previous findings that high cognitively demanding activities present difficulties for individuals with SCD, making IADLs a stronger predictor of SCD than ADLs [8,14,37]. Older adults with SCD experience less engagement in activities than those without SCD [8]. Our study showed various published articles assessing each IADL phenotype framed in the ICF [16]. Of particular interest, all eight published articles included domestic living activities in assessing SCD. Still, only one report used assessment of interpersonal interactions and relationships as a potential indicator of SCD. The task considered was relating to strangers [8]. Previous studies have shown that older people with more severe SCD reported more difficulties with the IADLs, specifically the social skill subdomain [7].

Our study findings showed that both ADLs and IADLs played a role in indicating SCD; thus, both categories strongly predict progression to dementia in patients in prodromal stages of AD [5,8]. Different cognitive domains in the neurodegenerative process manifest in functional performance and may explain the deficits in ADLs and IADLs. For example, a previous cross-sectional study investigated the link between neurocognitive domains and various aspects of daily living among 202 older people with mild AD [11]. The study found that cognitive memory and language were associated with IADLs, e.g., food preparation and driving, while ADLs, such as bathing and eating, were associated with attention [11]. These findings suggest that basic ADLs also require complex cognitive processes, particularly in individuals with mild AD.

The target population of this review is older people with depressive symptoms who experience SCD. As mentioned, depressive comorbidities may exacerbate deficits in everyday functioning. There are no clear lines discriminating the impact of SCD or depressive comorbidities on the risk of having a considerable degree of ADL impairment. Some of the published articles controlled for depression by applying specific statistical methods. For example, the raw scores of the B-ADL scale were transformed into z-scores [40] to adjust for depressive symptoms, where a B-ADL z-score < −1.5 indicates a considerable degree of functional impairment [5]. This statistical analysis minimizes the confounding effects of depressive symptoms on assessed ADL/IADL impairment. Thus, given the long predementia stage in AD [5], subtle functional changes that occur early in the spectrum of cognitive decline in individuals with depressive symptoms [7] should call for careful observation as they are at even higher risk for progression to Alzheimer’s disease.

Previous studies on cognitive assessment without much focus on ADL/IADL indicators may not be reliable in understanding the trajectory of impairment in later life [5]. They might lead to interventions being delivered without a solid theoretical framework [18]. ADL/IADL deficits may be a biomarker of AD-type neuropathological changes that occur years before distinct clinical symptoms, even in the stage of SCD [5], which could be a valuable guideline for clinicians and help refine MCI criteria.

### 4.1. Implications for Research and Practice

One suggested reason for the complex association between depressive symptoms and cognitive and functional performance was the degree of global impairment across the cognitive decline continuum (normal cognition-MCI-AD) [26]. Differing degrees of functioning or cognitive impairment may affect cognitive domains differently across cognitive domains: episodic memory, language, executive functions, and visuospatial abilities, along with the different cognitive function levels [26]. Knowledge of ADL/IADL subcategory phenotypes indicating SCD may drive individuals with MCI to seek assistance with more complex activities or IADLs, facilitating early recognition of cognitive impairment by caregivers or family members [37]. It is well-known that family members often notice daily functional deficits long before they are evident on psychometric tests in individuals demonstrating mild symptoms of cognitive decline [15]. Better characterization of prodromal stages of clinical signs will increasingly become an exciting research field and may be especially beneficial for developing future preventive therapeutic strategies.

### 4.2. Limitation of the Review 

Although we obtained very informative findings on subjective cognitive decline instrumental activities of daily living indicators, our review has several limitations: We limited our search to academic peer-reviewed journal articles published in the last 10 years in order to get the most up-to-date information about subjective cognitive decline and restricted activities of daily living. This may have caused us to miss relevant articles outside of this date range.

## Figures and Tables

**Figure 1 healthcare-10-01508-f001:**
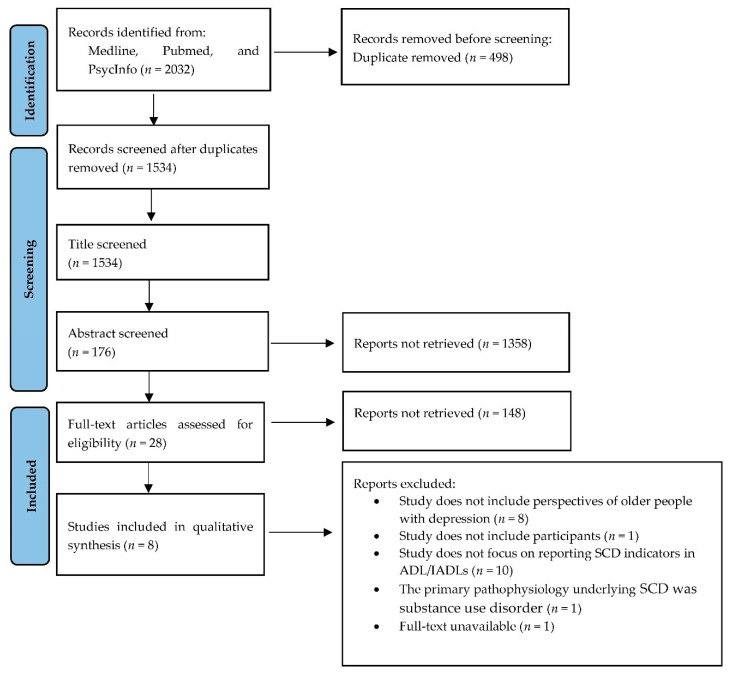
Flow diagram showing the numbers of publications identified and screened for eligibility during the scoping review.

**Table 1 healthcare-10-01508-t001:** Final search strategy on Medline via Ebsco Host.

	Key Concepts		Search Terms
	Activity of daily living	OR	“activities of daily living” OR ADL OR “occupational performance” OR IADL or “everyday functioning” OR “daily functioning” OR “activity performance” OR “daily life” OR “everyday life” OR “daily activities”
AND	Older adults	OR	“older adult” OR elderly OR geriatric * OR aging OR senior OR “older people” OR “aged 65” OR 65+ OR elderly OR senior * OR aged or older or elder or geriatric or “elderly people” OR “older people.”
AND	Subjective cognitive decline	OR	“subjective cognitive decline” OR “early dementia” OR “mild cognitive decline” OR “mild cognitive impairment” OR “age-associated cognitive impairment” OR “mild cognitive decline” OR “mild cognitive dysfunction”

* This symbol represents unlimited searches for variations on a word formed with different suffixes.

**Table 2 healthcare-10-01508-t002:** PCC framework of inclusion and exclusion criteria used in the literature search.

	Inclusion Criteria	Exclusion Criteria
Population Older adults with subjective cognitive decline (SCD)	Studies were included if they: Described the perspectives of older adults with SCD or MCI; Did not mention older adults explicitly, an older adult was defined as an adult aged 65 or above; Defined SCD as ‘a self-perceived condition where a person has experienced confusion or memory loss that is happening more often or is getting worse during the past 12 months.’	Focus exclusively on: Other age groups (e.g., children) or caregivers or health and social care professionals, or any other group; OR The primary pathophysiology underlying cognitive decline was outside the scope of age-related neurodegenerative processes, such as substance use disorder, traumatic brain injury, or stroke; The prevalence/incidence of a cognitive decline OR disease diagnosis/aetiology/clinical management
Concept Activities of daily living (ADLs) Instrumental activities of daily living (IADLs)	Studies were included if they: Discussed activities of older adults with a decline in cognitive function. ADLs were defined as core tasks of the everyday life of older adults, mainly to maintain personal care. IADLs were defined as more advanced tasks completed by older adults to maintain independent functioning. Described the lived experience of daily physical function of older adults with cognitive decline, OR Described impaired activities of the daily living of older adults with cognitive decline in an emergency or primary care or community.	Focus exclusively on: Financial needs such as housing benefits or pension credit; Development, evaluation, or assessment of interventions, services, or clinical tools.
Context Older adults with depressive symptoms	Studies were included if they: Assessed depressive or affective symptoms in older adults.	Focus exclusively on: Older adults’ experiences with ADLs/IADLs Determinants of health such as income, social status, education level, employment, genetics, gender, race, biomarkers.

**Table 3 healthcare-10-01508-t003:** Summary of the characteristics of the published articles included in the final analysis.

No.	Source	Location	Study Aim(s)	ADL/IADL Indicators under Investigation	Methods	Participant Characteristics
1.	(Reppermund et al., 2013) [8]	Sydney, Australia	Examine informant-based IADLs over two years in community-dwelling older individuals with MCI and explore whether the functional ability is predictive of cognitive decline.	ADLs and IADLs, measured on the Bayer-ADL scale.	A longitudinal study with a two-time point assessment. Participants aged 70–90 without dementia at baseline were recruited in the study center or their homes by trained psychologists. MCI diagnoses were based on subjective complaints, cognitive impairments in one domain or more, normal or minimally impaired functional ability, and no dementia. The Geriatric Depression Scale Depression assessed depression. Logistic regressions were used with diagnosis at baseline (MCI versus CN) and 2-year follow-up.	N: 602 Mean age (SD): 73.6–82.8 Female (%): 54.98 (331) Education: 7.98–15.3 years MCI: 227 Cognitively normal (CN): 375 Depression level in MCI group (SD): 2.19 (1.83) Depression level in the cognitively normal group (SD): 1.95 (1.71) ADL level in MCI group (SD): 1.60 (0.69) ADL level in CN group (SD): 1.37 (0.53)
2.	(Rovner et al., 2016) [9]	United States of America	Test the efficacy of a behavioral intervention to reduce cognitive decline by increasing activity participation in African American older adults.	IADL, measured on the Alzheimer’s Disease Cooperative Study Activities of Daily Living–Prevention Instrument (ADL-PI)	A cross-sectional randomized controlled trial. African American participants were recruited from the community and primary care clinics. The Geriatric Depression Scale assessed depression. The Hopkins Verbal Learning Test-Revised and the National Alzheimer’s Coordinating Center’s (NACC) Uniform Dataset Neuropsychological Battery Cognition assessed cognition. Statistical methods used multivariable regression, ANOVA, and Fisher’s Exact Test.	N: 221 Female (%): 80.1 (177) Mean age: 68.4–82.4 Mean education: 9.9–15.1 years Amnestic MCI-multiple domain (%): 79.6 (176) Amnestic MCI-single domain (%): 20.4 (45)
3.	(Stogmann et al., 2016) [5]	Vienna, Austria	Investigate ADL impairments in patients with SCD, MCI, and AD, compare them to healthy controls, and examine whether there is an association between impaired ADLs and depressive symptoms and neurocognitive functioning across different subgroups.	IADLs, measured on the Bayer Activities of Daily Living scale	Prospective cohort study. Participants were recruited from the memory outpatient clinic who complained of memory problems based on neurological examination, standard laboratory blood tests, and psychometric testing. Participants were excluded if having a stroke, a history of head injury, psychiatric diagnosis, or a medical condition leading to severe cognitive deterioration. Spearman’s correlation coefficients assessed the association between activities of daily living and other variables.	N: 1095 Median age: 60–77.5 Female (%): 53.94 Median for education: 8–15.5 Control: 343 SCD: 110 naMCI: 322 aMCI: 260 AD: 60
4.	(Burton et al., 2018) [18]	Canada	Investigate whether immediate memory, executive functions, depression, and apathy, predicted unique variance in IADL over and above demographic variables.	IADLs, measured on the Functional Activities Questionnaire	Cross-sectional cohort design. Participants were recruited from the memory clinic via referrals. Hierarchical regression was used to determine if executive function, delayed memory, depression, and apathy improve the prediction of IADLs.	N: 403 Age: 47.82–83.42 Female (%): 58.5 Years of education: 6.83–15.81 No cognitive impairment: 75 MCI: 75 Alzheimer’s disease: 139Non-AD dementia: 114
5.	(de Paula et al., 2016) [26]	Minas Gerais, Brazil	Evaluate how depressive symptoms moderate the cognitive and functional performance along a normal aging-MCI-AD continuum.	ADLs, measured on the Katz Index Independence ADL. IADLs, measured on the Lawton-Brody IADL scale.	Cross-sectional study with survey. Participants were recruited from community, voluntary, and statuary health and social care services for older adults with low vision or dementia. MCI was assessed based on Peterson’s (2001) criteria. MANOVA, logistic regression, and chi-square tests were performed to assess the association between depression and cognitive-functional performance in each group (NA × MCI × AD).	N: 274 Mean age: 64.85–81.64 Years of education: 0.71–9.51 NA non-depressed: 62 NA depressed: 34 MCI non-depressed: 63 MCI depressed: 22 AD non-depressed AD depressed: 27
6.	(Kim et al., 2020) [6]	South Korea	Identify the moderating effect of social support on the relationship between ADL and life satisfaction of older adults in both groups.	IADLs, measured on the Lawton-Brody Instrumental Activities of Daily Living Scale	Cross-sectional study with survey. Participants were recruited at 15 public health centers via convenience sampling. Older adults with severe cognitive impairment were excluded. Geriatric Depression Scale Short Form-Korea assessed depression. The Kolmogorov-Smirnov and Shapiro-Wilk tests and the Mann–Whitney U test compared the dementia high-risk group and low-risk group.	N: 609 Female (%): 65.85 Age: 60–80 and over Years of education: 0–13 and over The mean score for the Instrumental Activities of Daily Living Scale (IADL) was 52.71 (SD = 9.76) High-risk for dementia: 18.9% (113) Low-risk for dementia: 81.1 (496)
7.	(Yakhia et al., 2014) [24]	Nice, France	Evaluate levels of motor activity in MCI and healthy older populations and investigate the influence of depressive symptoms on mean motor activity in older people with MCI.	IADLs, as measured on recorded tasks such as making a phone call, preparing a pillbox in a specific order in 15 min under directed, semi-directed, and undirected conditions (ecological assessment)	Cross-sectional study with survey. Participants were recruited based on Petersen criteria for MCI diagnosis. Participants with a history of head trauma, loss of consciousness, aberrant motor behavior, and Parkinson’s disease symptoms were excluded. Neuropsychological tests assessed cognitive status. Montgomery-Asberg Depression Rating Scale, Geriatric Depression Scale, and NPI depression subscale assessed depression. Information and Communication Technologies assessed the motor activity of ten tasks. Mann-Whitney U-Test was used to compare the group with and without depression.	N: 36 Age: 67.14–80.28 Female (%): 44.44 (16) Years of education: 6–17 Control (%): 38.89 (14) MCI (%): 55.56 (20) MCI without depression (%): 10 (62.5) MCI with depression: 6 (37.5%)
8.	(Wu et al., 2021) [22]	Shanghai, China.	Examine the relationship between poor physical performance, malnutrition, depression, and cognitive impairment.	IADLs, as measured on the Lawton-Brody Independent Activities of Daily Living	Participants were older residents with initial symptoms of cognitive decline recruited from the community. The short physical performance battery assessed physical performance. The Chinese-version MMSE assessed cognition. Depressive symptoms were screened via the Chinese version of the Geriatric Depression Scale. The Mini Nutritional Assessment assessed nutrition status.	N: 1368 Age: 67.48–79.76 Female (%): 57.6 (809) MCI (%): 14.35 (199) Normal cognition (%): 86.767 (1187)

**Table 4 healthcare-10-01508-t004:** ADL/IADL indicators of subjective cognitive decline framed on ICF framework and coding system.

Activity and Participation	Learning & Applying Knowledge	General Tasks and Demands	Communication	Mobility	Self-Care	Domestic Life	Interpersonal Interactions and Relationships	Major Life Areas	Community, Social and Civic Life
Activities of daily living									
−d2301 Managing daily routine		[5,8]							
−d420 Transferring oneself				[24]					
−d450 Walking (going for a walk without getting lost)				[5,8]					
−510 Washing oneself (bathing)					[24]				
−d520 Caring for body parts (personal hygiene)					[5,8]				
−d530 Toileting					[24]				
−540 Dressing					[24]				
−d550 Eating					[24]				
−d598 Other specified self-care (continence)					[24]				
−d570 Looking after one’s health (medications, preparing a pillbox in a specific order in 15 min)									
Instrumental activities of daily living									
−d160 Focusing attention (paying attention to, understanding, discussing TV, book, magazine, concentrating on reading)	[5,8,18]								
−d163 Thinking	[18]								
−d170 Writing (keeping track of current events)	[18]								
−d179 Learning and applying knowledge (reading, writing)	[9]								
−d198 Other specified learning and applying knowledge	[5]								
−d2108 Undertaking single tasks, other specified (keeping track of current events)		[18]							
−d220 Undertaking multiple tasks (continuing after an interruption, doing two things at the same time, doing things safely)		[5,8]							
−d240 Handling stress and other psychological demands (performing a task under pressure)		[8]							
−d310 Communicating with receiving spoken messages (giving direction, taking a message for someone else)			[5,8]						
−d330 Speaking (describing what was heard, giving direction when asked, taking part in conversation)			[5,8]						
−d360 Using communication technology and devices (telephoning)			[22,24,26]						
−d460 Moving around in different locations (finding a way in an unfamiliar place)				[5,8]					
−d469 Walking and moving, other specified and unspecified (walking test, balance exercise, stand-up and go exercise, and a repeated rising from a chair to standing test—10 min)				[24]					
−d470 Using transportation				[5,6,7,8,18,22,26]					
−d475 Driving (traveling out of neighborhood, driving, arranging to take buses)				[18]					
−d499 Mobility, unspecified				[9]					
−d570 Looking after one’s health (medications, preparing a pillbox in a specific order in 15 min)						[6,9,22,24,26]			
−d6200 Acquisition of goods and service: shopping						[5,6,7,8,9,18,22,24,26]			
−d629 Acquisition of necessities, other specified and unspecified						[18]			
−d630 Household task (preparing meals or a balanced meal)						[5,6,7,8,9,18,22,24,26]			
−d640 Doing housework (housekeeping, laundry)						[7,18,22,26]			
−d6403 Using household appliances (heating water, turning off the stove after use)						[5,8,18]			
−d730 Relating with a stranger							[8]		
−d839 Education									
−d870 Economic self-sufficiency (writing cheques, paying bills, balancing a checkbook, counting money, personal finances (S), handling money^®^)								[5,7,9,18,22,26]	
−d8700 Economic resources (assembling tax records, business affairs, or papers)								[18]	
−d910 Community life (remembering appointments, family occasions, holidays, medications)									[8,18]
−d920 Recreation and leisure (hobby, participating in leisure)									[5,8]
−d998 Other specified community, social, and civic life									[8]

## Data Availability

Not applicable.

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
