# Peer review of "Daily Living Subjective Cognitive Decline Indicators in Older Adults with Depressive Symptoms: A Scoping Review and Categorization Using Classification of Functioning, Disability, and Health (ICF)"

_healthcare, 2022, doi:10.3390/healthcare10081508_

Round 1

Reviewer 1 Report

The topic is very relevant and provides with useful information and conclusions on the used indicators of SCD in ADL and IADLS. Furthermore, this article provides with a thorough literature review in the introduction. I only have minor remarks that can be used to provide with more clarity the readers of the paper. 

1) The authors use the term “successful aging” in page 4, without defining it. Are they using Rowe and Khan perspective? Or are they using it as an antonym from “depressive symptoms”?

 2)How was “depressive symptoms” addressed in the search strategy?

 3)Which is the rationale for using Medline, Pubmed and PsicINFO?

4)Which is the rationale for the time frame of 2011-2021?

 5)A section on limitations would be advisable.

Author Response

August 2, 2022

Dr. Dunja Stojanac

The Editor: Healthcare Journal

Dear Dr. Stojanac,

My research team and I are thankful for the constructive feedback we received from the Healthcare Journal review of our manuscript titled “Daily living subjective cognitive decline indicators in older adults with depressive symptoms: A scoping review and categorization using Classification of Functioning, Disability, and Health (ICF)”. We thank you for the opportunity to revise and resubmit the manuscript for further consideration by your esteemed journal, and believe it to have resulted in a much-improved manuscript from implementing the review recommendations.  

In our revisions, we carried out the changes implemented by the review panel and highlighted them in blue for easy viewing. As will be apparent from our revised draft, we made major revisions as advised by the review attached.

Sincerely,

The authors

Reviewer 2 Report

-The paper presents a review of daily living subjective cognitive decline indicators in older adults using the WHO International Classification of Functioning, Disability and Health (ICF).

-The review covers related published works in English language from January 2011 to November 2021.

-Data was analysed for 8 papers.

-The results reported by the authors indicated that eight articles provided evidence about the ADL (Activities of Daily Living) indicators of SCD (Subjective Cognitive Decline) in older adults with depressive symptoms.

-The authors concluded that highly cognitively demanding activities are more difficult to perform for individuals with SCD, and therefore IADLs (Instrumental Activities of Daily Living) are a stronger predictor of SCD than ADLs.

-I suggest removing the index numbers from the keywords and just include the keywords.

-The topic and area of research are relevant.

-The technical aspects of the research presented seem sound and detailed.

-Make sure that figures are clear to read and don't have misspellings. In Figure 1, the words displayed in blue boxes are not complete and there is a misspelling of the word "older".

Author Response

August 2, 2022

Dr. Dunja Stojanac

The Editor: Healthcare Journal

Dear Dr. Stojanac

My research team and I are thankful for the constructive feedback we received from the Healthcare Journal review of our manuscript titled “Daily living subjective cognitive decline indicators in older adults with depressive symptoms: A scoping review and categorization using Classification of Functioning, Disability, and Health (ICF)”. We thank you for the opportunity to revise and resubmit the manuscript for further consideration by your esteemed journal, and believe it to have resulted in a much-improved manuscript from implementing the review recommendations.  

In our revisions, we carried out the changes implemented by the review panel and highlighted them in blue for easy viewing. As will be apparent from our revised draft, we made major revisions as advised by the review attached.
